# Correlation Robust Influence Maximization

**Louis Chen**[*]
Naval Postgraduate School
Monterey, California

**Divya Padmanabhan**[†]
SUTD
Singapore

**Chee Chin Lim**[‡]
SUTD
Singapore

**Karthik Natarajan**[§]
SUTD
Singapore

## Abstract

We propose a distributionally robust model for the influence maximization problem. Unlike the classic independent cascade model [16], this model's diffusion process is adversarially adapted to the choice of seed set. Hence, instead of optimizing under the assumption that all influence relationships in the network are independent, we seek a seed set whose expected influence under the worst correlation, i.e. the "worst-case, expected influence", is maximized. We show that this worst-case influence can be efficiently computed, and though the optimization is NP-hard, a $(1 - 1/e)$ approximation guarantee holds. We also analyze the structure to the adversary's choice of diffusion process, and contrast with established models. Beyond the key computational advantages, we also highlight the extent to which the independence assumption may cost optimality, and provide insights from numerical experiments comparing the adversarial and independent cascade model.

## 1 Introduction

Social networks are models that capture transmission of information among its members. They find applications in testing effectiveness of policies, diffusion of medical innovations, marketing campaigns and news [34, 30, 13, 6]. Central to these models is a directed graph $G = (V, E)$ where $V$ denotes a set of nodes (members/users) and $E$ a set of edges (influence relationships). In progressive diffusion models, a subset of the edges are randomly deemed "live" via a binary valued random vector $\tilde{c}$ of size $|E|$, in which $\tilde{c}_{ij} = 1$ (w.p. $p_{ij}$) iff edge $(i, j)$ is "live," so that given a seed set $\mathcal{S} \subseteq V$, all nodes reachable along live-edge paths from $\mathcal{S}$ (including $\mathcal{S}$ itself) are activated, or "influenced." The *influence maximization* problem is thus to find a $k$-sized seed set $\mathcal{S} \subseteq V$ that maximizes the number $R(\tilde{c}, \mathcal{S})$ of influenced nodes in expectation, i.e., $\max_{|\mathcal{S}| \leq k} \mathbb{E}[R(\tilde{c}, \mathcal{S})]$, where $k$ represents, say, a viral marketing campaign's budget constraint.

There have been many studies to model diffusion and the spread of influence in graphs ([7, 21, 32]). In particular, [16] was seminal in studying influence maximization under the well-known *Independent Cascade (IC)* model, in which all edges are independently live - equivalently, the components to $\tilde{c}$ are mutually independent Bernoulli random variables and we write $\tilde{c} \sim \theta_{ic}$. The IC influence maximization problem - $\max_{|\mathcal{S}| \leq k} \mathbb{E}_{\theta_{ic}}[R(\tilde{c}, \mathcal{S})]$ - is known to be NP-hard; in fact, even evaluating $f^{ic}(\mathcal{S}) := \mathbb{E}_{\theta_{ic}}[R(\tilde{c}, \mathcal{S})]$ for a seed set $\mathcal{S}$ is #P-hard [16], though several works have proposed efficient approximation methods, and a greedy algorithm provides a $1 - 1/e - \epsilon$ approximation guarantee[16], where $\epsilon > 0$ accounts for sampling errors involved in the approximation of $f^{ic}(\mathcal{S})$. But beyond the computation, the independence assumption itself may not be appropriate. Indeed, while a graph may capture observable connections in a social network with edge connectivity describing friends, followers, etc., there may be latent variables causing apparently disconnected segments of

---

[*]louis.chen@nps.edu

[†]divya_padmanabhan@sutd.edu.sg

[‡]cheechin_lim@sutd.edu.sg

[§]karthik_natarajan@sutd.edu.sg

a network to in fact display correlated capacities to propagate influence. And for that matter, the correlation could depend on the particular idea, product, or news to be propagated.

In this work, we assume knowledge of the live-edge probabilities $p_{ij}$ but make no assumptions on the correlation(/dependence) of all $\tilde{c}_{ij}$. Thus all joint distributions consistent with the marginal probabilities $p_{ij}$ are admissible. And we propose to choose a k-sized seed set $\mathcal{S}$ that maximizes the expected number of influenced nodes with respect to the worst correlation. In such an adversarial model we hedge against unknown dependence structure to the random live edges, or equivalently identify a set of nodes that are influential regardless of the model.

The techniques used in this work fall under the realm of distributionally robust optimization (DRO), a research area concerned with optimizing a risk measure, under the most adverse member from a family of distributions. By introducing such an adversarial selection, decision-makers now seek decisions that would be "robust" to assuming an incorrect distribution. This approach has gained in popularity in several domains such as machine learning and game theory[10, 28, 18]. The family of distributions we consider in this work is commonly referred to as the Fréchet class (all joint distributions consistent with the given marginals) in the literature [22, 17, 5].

**Contributions**:

(1) We first formulate a distributionally robust model for influence maximization in Section 3. Unlike the IC model, this adversarial model allows for a polynomial time solvable linear programming formulation to obtain the worst case expected influence function, given a seed set $\mathcal{S}$. The linear program contains $|V|$ variables and $O(|E| + |V|)$ constraints and is interpretable.

(2) In Section 4, we show that finding an optimal seed set $\mathcal{S}$ that maximizes the worst case expected value $R(\tilde{\mathbf{c}}, \mathcal{S})$ is NP-hard. Interestingly, despite the adversarial nature of the model, we show that the worst case expected influence function is submodular in $\mathcal{S}$. By using the greedy algorithm, we obtain an approximation guarantee of $(1 - 1/e)$, with no need to resort to simulation methods.

(3) We establish that if the activation events are not necessarily independent, operating under the assumption of independence can hurt the expected influence. We quantify this notion by adapting the price of correlations (POC) ratio [2] to the context of influence maximization.

(4) An experimental study of using a distributionally robust model on real world datasets is provided in Section 5. The key benefit offered by a distributionally robust model is computational efficiency and the robustness of the generated seed set to the dependence among the edges.

## 2   Related Work and Preliminaries

There is an extensive literature on influence maximization, including adaptive models [27], learning (e.g. [25, 12, 4]), and in recent years, robustness. To the best of our knowledge, robustness in influence maximization first received attention through the parametric perturbation interval model [11] where for each edge $(i, j) \in E$ the probability $p_{ij}$ is not known exactly, but rather lies in an interval $[l_{ij}, r_{ij}] \subseteq [0, 1]$. The model however still assumes all edges are independently live. Their objective is to obtain the best seed set under the IC model, robust to the values the edge likelihoods $\mathbf{p}$ can take. Models of a similar spirit include [8, 15, 14, 29]. Additionally, [8, 14] study robustness from the view of model misspecification; the particular objectives studied there are hard, partly due to their non-convex and non-submodular objectives. An analogous study of robustness has been performed with the linear threshold model (another diffusion process) in [24] where the parameters are assumed to be uncertain.

While the majority of robust studies for the IC model have considered parameter uncertainty by way of the edge likelihoods, and hence still assume a fixed correlation structure - namely, independent edge propagation, we study the "reverse'" problem by assuming the edge likelihoods are fixed and the uncertainty lies in how they are correlated. Indeed, edge likelihoods are amenable to estimation individually while estimation of multivariate joint distributions is generally intractable. There has been prior interest in modeling the contribution that correlations may play [31, 33, 3]. In particular, [31] replace the influence function with a surrogate function that provides the most optimistic expected spread of influence. This, too, is a reversal of this work's focus - a consideration of the *pessimistic* expected spread of influence.

## 2.1 Preliminaries

While influence maximization employs a stochastic setting, it will be useful to first consider a deterministic one, which can be characterized in an intuitive way. Given a graph $G = (V, E)$, a set of seed nodes $\mathcal{S} \subseteq V$, and a vector $\mathbf{c}$ containing $|E|$ binary components, if we add two distinct nodes $s$ and $t$ along with accompanying incident edges to form the auxiliary graph $G' := (V \cup \{s, t\}, E')$, where $E' := \{(i, j) \in E : c_{ij} = 1\} \cup \{(s, i) : i \in \mathcal{S}\} \cup \{(j, t) : j \in V \setminus \mathcal{S}\}$, then the following max flow problem on $G'$ computes $R(\mathbf{c}, \mathcal{S})$.

**Lemma 1** *Let $Z(\mathbf{c}, \mathcal{S})$ denote the value to the following max flow problem on the graph $G' := (V \cup \{s, t\}, E')$*

$$Z(\mathbf{c}, \mathcal{S}) = \max_{x \in \mathbb{R}^{E'}, v \in \mathbb{R}} v$$

$$s.t. \sum_{j:(i,j) \in E'} x_{ij} - \sum_{j:(j,i) \in E'} x_{ji} = \begin{cases} v, & i = s \\ 0, & i \in V \\ -v, & i = t \end{cases}$$

$$x_{jt} \leq 1 \ \ for \ j \in V \setminus \mathcal{S}, x_{ij} \geq 0 \ \ for \ (i, j) \in E',$$

*and $x^*$ denote an optimal flow. Then, the number of nodes reachable from $\mathcal{S}$ along the live edges specified in $\mathbf{c}$ is $R(\mathbf{c}, \mathcal{S}) = |\mathcal{S}| + Z(\mathbf{c}, \mathcal{S})$.*

The max flow problem of $Z(\mathbf{c}, \mathcal{S})$ provides a natural way to describe $R(\mathbf{c}, \mathcal{S})$. Indeed, the members of $j \in V \setminus \mathcal{S}$ such that in the optimal flow, $x_{jt}^* = 1$, are the nodes that are reachable from $s$ in the graph $G'$- equivalently, they are the nodes reachable from $\mathcal{S}$ in the graph $G$ along only the edges in $E$ such that $c_{ij} = 1$. An example of the max flow problem in Lemma 1 is illustrated in Figure 1. The bold edges denote those members $(i, j) \in E$ such that $c_{ij} = 1$.

## 3 The Correlation Robust Influence Function

Let $\mathbf{p} = (p_e)_{e \in E}$ be a vector of edge likelihoods. $\mathbf{p}$ is a vector of size $|E|$ where $p_e$ (equivalently $p_{ij}$) denotes the activation probability of edge $e = (i, j) \in E$ and $0 \leq p_e \leq 1$. Suppose $\mathcal{C}$ denotes the set of all binary vectors of size $|E|$ and let,

$$\Theta = \{\theta \in \mathbb{R}_+^{\mathcal{C}} : \sum_{\mathbf{c} \in \mathcal{C}: c_{ij} = 1} \theta(\mathbf{c}) = p_{ij} \ \ \forall (i, j) \in E, \sum_{\mathbf{c} \in \mathcal{C}} \theta(\mathbf{c}) = 1\}$$

denote the set of all joint probability distributions over $\mathcal{C}$ consistent with the marginals provided by $\mathbf{p}$ (we suppress the dependency for brevity). $\theta$ is a member distribution over the set $\mathcal{C}$ of $2^{|E|}$ graph realizations. $\theta(\mathbf{c})$ denotes the probability mass assigned to any particular realization $\mathbf{c}$. $\Theta$ is non-empty (as $\theta_{ic} \in \Theta$), and uncountable in general. We define the *correlation robust influence function* as follows,

$$f^{corr}(\mathcal{S}) := \min_{\theta \in \Theta} \mathbb{E}_\theta[R(\tilde{\mathbf{c}}, \mathcal{S})] \ \ \forall \mathcal{S} \subseteq V, \quad \text{(Correlation Robust Influence Function)} \quad (1)$$

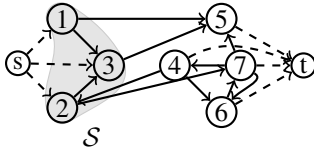

Figure 1: Construction of the auxiliary graph $G'$

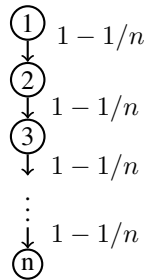

Figure 2: Example 1 for POC study

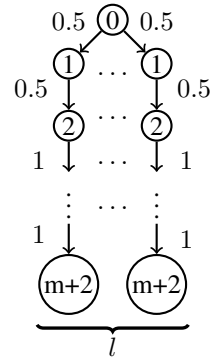

Figure 3: Example 2 for POC study

which, for any seed set $\mathcal{S}$, yields the worst case expected number of influenced nodes among all consistent joint distributions (and in turn over all possible correlation structures). We formulate the *correlation robust influence maximization problem* as:

$$\max_{\mathcal{S}:|\mathcal{S}|\leq k} f^{corr}(\mathcal{S}) \tag{2}$$

This models the problem of finding a set of nodes that are influential regardless of the correlation structure, by maximizing the worst-case expected influence. Lemma 1 gives a way to formulate the computation of $f^{corr}(\mathcal{S})$, since $f^{corr}(\mathcal{S}) = \min_{\theta \in \Theta} \mathbb{E}_\theta[R(\tilde{\mathbf{c}}, \mathcal{S})] = |\mathcal{S}| + \min_{\theta \in \Theta} \mathbb{E}_\theta[Z(\tilde{\mathbf{c}}, \mathcal{S})]$. The expected number of influenced members outside the seed set, $\min_{\theta \in \Theta} \mathbb{E}_\theta[Z(\tilde{\mathbf{c}}, \mathcal{S})]$, amounts to an instance of the distributionally robust max flow problem studied in [5]. Using one of the main results therein, we derive the following efficient linear program representation.

**Theorem 1** *Let $G = (V, E)$ be a directed graph, $\mathcal{S} \subseteq V$ a seed set, and $\mathbf{p} \in [0, 1]^E$ a vector of edge likelihoods. Then $\min_{\theta \in \Theta} \mathbb{E}_\theta[Z(\tilde{\mathbf{c}}, \mathcal{S})]$ is the value of the following polynomial sized linear program.*

$$
\begin{aligned}
\min_{\theta \in \Theta} \mathbb{E}_\theta[Z(\tilde{\mathbf{c}}, \mathcal{S})] = \min_{\boldsymbol{\pi} \in \mathbb{R}^V} \quad & \sum_{i \in V \setminus \mathcal{S}} \pi_i \\
s.t \quad & \pi_i = 1 \ \text{for } i \in \mathcal{S}, \\
& \pi_i - \pi_j \leq 1 - p_{ij} \ \text{for } (i,j) \in E, \\
& 0 \leq \pi_i \leq 1 \ \text{for } i \in V
\end{aligned} \tag{3}
$$

We remark that Theorem 1 implies that, for any seed set $\mathcal{S}$, $f^{corr}(\mathcal{S})$ is efficiently computable with linear programming, in contrast to $f^{ic}(\mathcal{S}) = \mathbb{E}_{\theta_{ic}}[Z(\tilde{\mathbf{c}}, \mathcal{S})]$, which is NP-hard to even approximate [16]. But, further, with the form of (3) in view, we may also speak on the *correlation robust likelihood* that any node $i \notin \mathcal{S}$ will be influenced. This is the topic of the next result, which we take a moment to motivate.

Consider any distribution $\theta \in \Theta$ and suppose $\tilde{\mathbf{c}} \sim \theta$. Let $G(\tilde{\mathbf{c}}) = (V, E(\tilde{\mathbf{c}}))$ be the random live-edge graph induced by $\tilde{\mathbf{c}}$ in which $E(\tilde{\mathbf{c}}) := \{(i,j) \in E : \tilde{c}_{ij} = 1\}$. In the graph $G(\tilde{\mathbf{c}})$, a node $i$ is influenced by a node in $\mathcal{S}$ if and only if there exists a directed path $\gamma = (i_0 \to i_1 \to i_2 \to \ldots \to i_\lambda = i)$ of some positive length $\lambda_\gamma$ from a node $i_0 \in \mathcal{S}$ to $i$. This implies that for any $\theta \in \Theta$,

$$\mathbb{P}_\theta(G(\tilde{\mathbf{c}}) \text{ contains path } \gamma) = 1 - \mathbb{P}_\theta(\cup_{l=0}^{\lambda_\gamma} [(i_{l-1} \to i_l) \notin E(\tilde{\mathbf{c}})]) \geq 1 - \sum_{l=0}^{\lambda_\gamma} (1 - p_{i_{l-1}, i_l}), \quad (4)$$

by the union bound. If we denote the collection of all such directed paths from $\mathcal{S}$ to $i$ in the original graph $G$ as $\Gamma(\mathcal{S}, i)$, and let $L(\gamma) := 1 - \sum_{l=0}^{\lambda_\gamma} (1 - p_{i_{l-1}, i_l})$ for any $\gamma \in \Gamma(\mathcal{S}, i)$, then we get the following lower bound on the influence likelihood for any $\theta$.

$$\mathbb{P}_\theta(\text{Node } i \text{ is reachable from } \mathcal{S} \text{ in } G(\tilde{\mathbf{c}})) = \mathbb{P}_\theta(\cup_{\gamma \in \Gamma(\mathcal{S}, i)} G(\tilde{\mathbf{c}}) \text{ contains path } \gamma) \geq [\max_{\gamma \in \Gamma(\mathcal{S}, i)} L(\gamma)]^+ \tag{5}$$

$$
\begin{aligned}
\min_{\theta \in \Theta} \mathbb{E}_\theta[Z(\tilde{\mathbf{c}}, \mathcal{S})] &= \min_{\theta \in \Theta} \sum_{i \in V \setminus \mathcal{S}} \mathbb{P}_\theta(\text{Node } i \text{ is reachable from } \mathcal{S} \text{ in } G(\tilde{\mathbf{c}})) \\
&\geq \sum_{i \in V \setminus \mathcal{S}} \min_{\theta \in \Theta} \mathbb{P}_\theta(\text{Node } i \text{ is reachable from } \mathcal{S} \text{ in } G(\tilde{\mathbf{c}})) \geq \sum_{i \in V \setminus \mathcal{S}} [\max_{\gamma \in \Gamma(\mathcal{S}, i)} L(\gamma)]^+
\end{aligned} \tag{6}
$$

The following corollary shows that all the inequalities in (5) and (6) turn out to be equalities.

**Corollary 1 (Correlation Robust Influence Likelihood)** *For an arbitrary seed set $\mathcal{S}$ and vector of edge likelihoods $\mathbf{p} \in [0, 1]^E$, let $\pi^*$ solve (3). Then for each $i \in V \setminus \mathcal{S}$,*

$$\pi_i^* = [\max_{\gamma \in \Gamma(S, i)} L(\gamma)]^+, \tag{7}$$

$$\mathbb{P}_{\theta^*}(\text{Node } i \text{ is reachable from } \mathcal{S} \text{ in } G(\tilde{\mathbf{c}})) = \pi_i^* \qquad \forall \theta^* \in \underset{\theta \in \Theta}{\arg\min} \, \mathbb{E}_{\tilde{\mathbf{c}} \sim \theta}[Z(\tilde{\mathbf{c}}, \mathcal{S})].$$

*In light of this, we define the **correlation robust influence likelihood of** $i$ as $\pi_i^*$. In particular, $\pi_i^* \leq \mathbb{P}_{\theta_{ic}}(\text{Node } i \text{ is reachable from } \mathcal{S}).$*

Corollary 1 leads to the interesting contrast of the form (7) with the IC model's likelihood of $1 - \Pi_{l=0}^{\lambda_\gamma}(1 - p_{i_{l-1},i_l})$. Compared to the IC model likelihood, $\pi_i^*$ and in turn $f^{corr}(\cdot)$ degrades at a faster rate along a path due to the form of the likelihood. As a consequence, we expect a seed set that maximizes correlation robust influence $f^{corr}(\cdot)$ will in general be "spread out" in comparison to the one that maximizes $f^{ic}(\cdot)$, especially when the edge likelihoods are small. The next result concerns the structure of the random graph $G(\tilde{\mathbf{c}})$. In contrast to the IC model, under the correlation robust coupling $\theta^*$, not all paths in $\Gamma(\mathcal{S},i)$ contribute to the likelihood that a node $i \notin \mathcal{S}$ is influenced. In truth, only a subset of these paths ever manifest (and always appear together) with positive probability.

**Corollary 2 (Path existence under Correlation Robustness)** *Let $\mathcal{S}$ be an arbitrary seed set, and let $\theta^* \in \Theta$ be any solution to $\min_{\theta \in \Theta} \mathbb{E}_\theta[R(\tilde{\mathbf{c}}, \mathcal{S})]$. Let $\bar{\Gamma}(\mathcal{S},i) := \arg\max_{\gamma \in \Gamma(\mathcal{S},i)} L(\gamma)$, and $\pi^*$ is the optimal solution to (3). If $i \notin \mathcal{S}$ and $\max_{\gamma \in \Gamma(S,i)} L(\gamma) > 0$, then*

$$\mathbb{P}_{\theta^*}(\cup_{\gamma \in \bar{\Gamma}(\mathcal{S},i)} [G(\tilde{\mathbf{c}}) \text{ contains path } \gamma]) = \pi_i^* = \mathbb{P}_{\theta^*}(\cap_{\gamma \in \bar{\Gamma}(\mathcal{S},i)} [G(\tilde{\mathbf{c}}) \text{ contains path } \gamma]),$$

*In addition, if $\max_{\gamma \in \Gamma(S,i)} L(\gamma) > 0$, then for any path $\gamma \in \bar{\Gamma}(\mathcal{S},i)$ and $\tilde{\mathbf{c}} \sim \theta^*$, at most one of the arcs in $\gamma$ is ever missing in the random graph $G(\tilde{\mathbf{c}})$ almost surely.*

We now characterize the adversarial distribution. This provides a way to simulate the resulting influence, and allows for an interesting comparison to the linear threshold model (LTM) [16].

**Corollary 3** *Given an arbitrary seed set $\mathcal{S}$ and vector of edge likelihoods $\mathbf{p} \in [0,1]^E$, let $\pi^*$ denote the optimal solution to (3). Let $\tilde{q} \sim Unif[0,1]$, $V(\tilde{q}) := \{i : \tilde{q} < \pi_i^*\}$,*

$$E(\tilde{q}) := \{(k,j) : \pi_k^* > \pi_j^*, \tilde{q} \notin [\pi_k^* - 1 + p_{kj}, \pi_k^*]\} \cup \{(k,j) : \pi_k^* \leq \pi_j^*, \tilde{q} \in (0, p_{kj}]\},$$

*and $\mathbf{c}(\tilde{q}) \in \{0,1\}^E$ be such that $c(\tilde{q})_{ij} = 1$ iff $(i,j) \in E(\tilde{q})$. Then $\mathbf{c}(\tilde{q}) \sim \theta^*$ for some $\theta^*$ solving Equation (1). In particular, $V(\tilde{q})$ is the set of all nodes reachable from $\mathcal{S}$ in the graph $G(\tilde{q}) = (V, E(\tilde{q}))$, so that $\mathbb{E}_{\tilde{q}}[|V(\tilde{q})|] = \min_{\theta \in \Theta} \mathbb{E}_{\tilde{\mathbf{c}} \sim \theta}[R(\tilde{\mathbf{c}}, \mathcal{S})] = |\mathcal{S}| + \mathbb{E}_{\tilde{q}}[Z(\mathbf{c}(\tilde{q}), \mathcal{S})]$.*

Corollary 3 characterizes both the random set of influenced nodes $V(\tilde{q})$ and the random "live edges" $E(\tilde{q})$ that allow $\mathcal{S}$ to reach them using a single random number $\tilde{q} \in [0,1]$. This is in contrast to LTM in which such a draw from $\text{Unif}[0,1]$ is required for every node. Further, in LTM at most one live edge enters any node $i$, while under correlation robustness either all the paths in $\bar{\Gamma}(\mathcal{S},i)$ are live simultaneously or $i$ is not reached at all (from Corollary 2).

## 4   Correlation Robustness: Maximization and Price of Correlations

### 4.1   The Correlation Robust Influence Maximization Problem

We will now investigate the problem of computing the best set of seed nodes that maximizes $f^{corr}(\cdot)$.

**Theorem 2** *The problem of computing $\max_{\mathcal{S}:|\mathcal{S}|\leq k} f^{corr}(\mathcal{S})$, given a graph $G = (V,E)$, a vector of edge likelihoods $\mathbf{p} \in [0,1]^E$, and an integer number $k$, is NP-Hard. In particular, we have the following exact formulation as a mixed-integer program (MILP).*

$$\max_{\mathcal{S}:|\mathcal{S}|\leq k} f^{corr}(\mathcal{S}) = \max_{\mathbf{x},\mathbf{y},\mathbf{w},\mathbf{q},\mathbf{z}} \sum_{(i,j)\in E} z_{ij}(p_{ij} - 1) + \sum_{i \in V} w_i - q_i$$

$$1 - y_i - \sum_{j:(j,i)\in E} z_{ji} + \sum_{j:(i,j)\in E} z_{ij} + q_i \geq 0 \ \forall i \in V$$

$$|V|x_i + y_i - |V| \leq w_i \leq \min(|V|x_i, y_i) \ \forall i \in V$$

$$\sum_{i \in V} x_i = k, x_i \in \{0,1\} \ \forall i \in V$$

$$y_i \geq 0, q_i \geq 0 \ \forall i \in V, z_{ij} \geq 0 \ \forall (i,j) \in E$$

*Let $\mathbf{x}^*, \mathbf{y}^*, \mathbf{w}^*, \mathbf{q}^*, \mathbf{z}^*$ be the optimal solution to the MILP. The optimal seed set $\mathcal{S}_{corr} = \{i : x_i^* = 1\}$.*

Next we show that the function $f^{corr}(\cdot)$ is a submodular function. If $g(\cdot)$ and $h(\cdot)$ are submodular set functions, the pointwise minimum $\min(g(\cdot), h(\cdot))$ need not be a submodular function in general (see the book on submodular functions by Bach, 2013). But for $f^{corr}(\mathcal{S}) = \min_{\theta \in \Theta} \mathbb{E}_\theta[R(\tilde{\mathbf{c}}, \mathcal{S})]$ as a pointwise minimum of submodular functions $\mathbb{E}_\theta[R(\tilde{\mathbf{c}}, \mathcal{S})]$ over $\theta \in \Theta$, it turns out to be submodular.

**Theorem 3** *The correlation robust influence function $f^{corr}(\cdot)$ is a monotone, submodular function.*

A consequence of the above two theorems is that though the correlation robust maximization problem is NP-hard, a greedy algorithm for maximizing $f^{corr}(\cdot)$ carries desirable guarantees. The greedy algorithm terminates after $k$ steps. The initial seed set $\mathcal{S}^{(0)} = \emptyset$. At iteration $i$, the seed set $\mathcal{S}^{(i)} = \mathcal{S}^{(i-1)} \cup v^{(i)}$, for any $v^{(i)} \in \arg\max_{v \in V \setminus \mathcal{S}^{(i-1)}} f^{corr}(\mathcal{S}^{(i-1)} \cup \{v\})$, and the output upon termination is $\mathcal{S}^g_{corr} = \mathcal{S}^{(k)}$.

**Corollary 4** *Let $\mathcal{S}^g_{corr}$ denote the seed set generated upon termination of the greedy algorithm for maximization of $f^{corr}(\cdot)$. Then, $f^{corr}(\mathcal{S}^g_{corr}) \geq (1 - 1/e)\max_{|\mathcal{S}| \leq k} f^{corr}(\mathcal{S})$.*

Note the departure from IC, where the approximation guarantee of the greedy algorithm is $(1 - 1/e - \epsilon)$, $\epsilon$ the result of sampling error in estimation of $f^{ic}(\mathcal{S})$ [16]. Under correlation-robustness, the sampling error does not appear, as $f^{corr}(\cdot)$ is polynomial time computable with linear programming.

### 4.2 Price of Correlations and Correlation Gap

In this section, we examine the extent to which assuming independence could cost the decision maker. If a decision maker assumes independence and uses a seed set $\mathcal{S}_{ic} \in \arg\max_{\mathcal{S}:|\mathcal{S}|\leq k} f^{ic}(\mathcal{S})$, as opposed to any $\mathcal{S}_{corr} \in \arg\max_{\mathcal{S}:|\mathcal{S}|\leq k} f^{corr}(\mathcal{S})$, then the *price of correlations* (POC) (coined in [2]) characterizes the suboptimality that $\mathcal{S}_{ic}$ presents in the optimization of $f^{corr}(\cdot)$, with the ratio

$$\text{POC} = \frac{f^{corr}(\mathcal{S}_{ic})}{\max_{\mathcal{S}:|\mathcal{S}|\leq k} f^{corr}(\mathcal{S})} \qquad \text{(Price of Correlations)}$$

Intuitively, POC describes the cost of using a seed set optimal to an 'incorrect' diffusion model. For our influence maximization setting, we also define $\kappa(\mathcal{S})$ (known as *Correlation Gap* in [2]) as, $\kappa(\mathcal{S}) = \frac{f^{corr}(\mathcal{S})}{f^{ic}(\mathcal{S})}$. Since, $\max_{\mathcal{S}:|\mathcal{S}|\leq k} f^{corr}(\mathcal{S}) = f^{corr}(\mathcal{S}_{corr}) \leq f^{ic}(\mathcal{S}_{corr}) \leq f^{ic}(\mathcal{S}_{ic})$, we have,

$$1 \geq \text{POC} = \frac{f^{corr}(\mathcal{S}_{ic})}{\max_{\mathcal{S}:|\mathcal{S}|\leq k} f^{corr}(\mathcal{S})} \geq \frac{f^{corr}(\mathcal{S}_{ic})}{f^{ic}(\mathcal{S}_{ic})} = \kappa(\mathcal{S}_{ic}) \geq 0 \qquad (8)$$

A POC value closer to 1 indicates that we do not suffer much by resorting to $\mathcal{S}_{ic}$ when the underlying diffusion process corresponds to an adversarial model. But a POC value close to zero indicates major loss. We demonstrate that both of these scenarios are certainly possible under the right graph structure.

**Example 1:** We first consider the series graph in Figure 2 where $n \geq 2$. Let $p_{ij} = 1 - 1/n$ for all the edges and let $k = 1$. It can be verified by inspection that $\mathcal{S}_{corr} = \mathcal{S}_{ic} = \{1\}$. Then, the correlation gap $\kappa(\mathcal{S}_{ic})$ can be computed as,

$$\kappa(\mathcal{S}_{ic}) = \frac{f^{corr}(\mathcal{S}_{ic})}{f^{ic}(\mathcal{S}_{ic})} = \frac{1 + \sum_{i=2}^{n} 1 - \frac{i-1}{n}}{1 + \sum_{i=1}^{n-1}(1 - 1/n)^i} = \frac{1 + (n-1)/2}{n\left[1 - (1 - 1/n)^n\right]}$$

Therefore $\lim_{n\to\infty} \kappa(\mathcal{S}_{ic}) = 1/2 \cdot e/(e-1) \approx 0.791$ and from Equation (8), the price of correlations is at least $0.791$ asymptotically.

**Example 2:** We next consider the tree in Figure 3 where the root node contains $l$ children. There are a total of $l$ paths from the root to all the leaf nodes. Each path contains $m + 2$ nodes (apart from the root). The labels on the nodes depict the "type" of each node. Edges between nodes of type 0 and 1 as well as between type 1 and type 2 nodes have $0.5$ probability of being live. For all other edges, the probability is 1. The total number of nodes in the graph is $n = l(m + 2) + 1$, where we let $\frac{4m}{m+3} \leq l \leq 2m$. Then if $k = 1$, $\mathcal{S}_{corr}$ is any one of the type 2 nodes while $\mathcal{S}_{ic} = \{0\}$ (details in the supplementary material). Thus POC $= ((l/2) + 1)/(m + 1)$. If $l = 4m/(m + 3)$, POC $= (2m + 3)/((m + 1)(m + 3))$ which tends to zero as $m \to \infty$. This example leads to the following theorem.

**Theorem 4** *There exists a graph on $n$ nodes with price of correlations $O(1/n)$.*

The theorem reveals that in general, the POC may be arbitrarily close to zero. In other words, $\mathcal{S}_{ic}$ can be arbitrarily sub-optimal, if used under an adversarial diffusion process.

# 5 Experiments

We now discuss experiments comparing the IC and correlation robust models [5].

**Datasets** Our experiments were performed on two datasets (1) `wikivote`: Here each node denotes a user and each edge $(i, j)$ denotes the action of user $i$ voting for user $j$ to be an admin [20]. As in [35, 36] we reverse the edges so that, edge $(i, j)$ in the original graph becomes $(j, i)$. Indeed, this reverse direction more aptly captures a notion of influence, as user $i$'s vote for $j$ establishes that user $j$ has influence over user $i$. (2) `polblogs`: Each node denotes a blog and each edge $(i, j)$ denotes that blog $i$ references blog $j$ via a hyperlink [1]. Since a highly referenced blog is "influential". Just as in `wikivote`, we reversed the direction of all edges. Table 1 provides summaries of the datasets.

| Dataset | $|V|$ | $|E|$ | Min Deg | Average Deg | Max Deg |
|---|---|---|---|---|---|
| `wikivote` | 7115 | 103689 | 1 | 29.146 | 1167 |
| `polblogs` | 1490 | 19022 | 0 | 25.532 | 467 |

Table 1: Numerical Summaries of the Graphs from the Datasets

**Implementation:** From Corollary 1, it follows that $f^{corr}(\cdot)$ can be computed in terms of equivalent shortest path calculations. In particular, the edge weights for the equivalent shortest path calculations are $1 - p_{ij}$ (see (4) and (5)). We used the `igraph` Python library [9] to represent the graphs and for the shortest path calculations. For computation of $f^{ic}(\cdot)$, we performed pruned Monte Carlo simulations [26]. Each set of Monte Carlo simulations comprised 10000 runs each, and we report an average over 10 such sets. We used an ASUS laptop with i7-7500U processor for all experiments.

**Computational Times**: We first illustrate the computational times for two instances of identical edge probabilities in Figure 4 as a function of the size of the seed set $k$. Since the MILP in Theorem 2 was found to be computationally expensive for the datasets described, we work with $\mathcal{S}^g_{corr}$. To obtain $\mathcal{S}^g_{ic}$ and $\mathcal{S}^g_{corr}$ (the greedy maximizers of $f^{ic}(.)$ and $f^{corr}(.)$), we used the accelerated greedy technique [19], a method used to maximize any monotone submodular function in a greedy manner. The accelerated greedy algorithm [23, 19] is designed to produce the greedy maximizers, albeit with reductions in the number of computations performed while searching for the node with the highest marginal gain. Figure 4 provides the times for the computation of $\mathcal{S}^g_{ic}$ and $\mathcal{S}^g_{corr}$. For IC, the error bars over 10 runs are provided. The computational times do not increase much with $k$, due to the use of the accelerated greedy algorithm. The plots show the computational advantage offered by the worst case model. Computation of $\mathcal{S}^g_{ic}$ is much more expensive than $\mathcal{S}^g_{corr}$. For instance, in Figure 4b, $\mathcal{S}^g_{corr}$ for the larger wikivote dataset can be computed in less than $100s$ while $\mathcal{S}^g_{ic}$ takes at least $400s$.

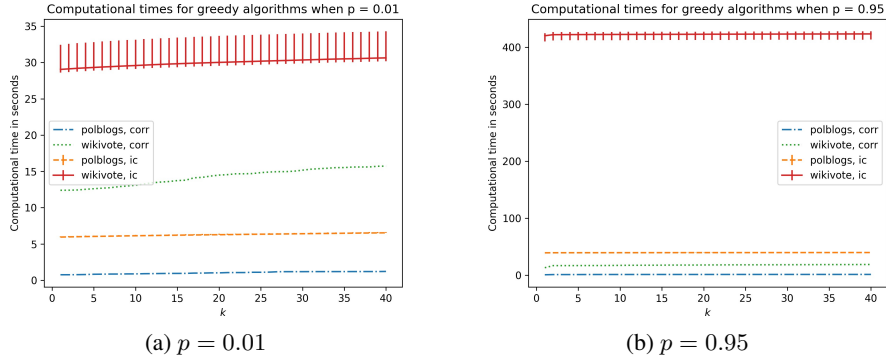

(a) $p = 0.01$          (b) $p = 0.95$

Figure 4: Plot of computational times, 'corr' and 'ic' refer to the times for obtaining $\mathcal{S}^g_{corr}$ and $\mathcal{S}^g_{ic}$.

**Properties of Seed Sets**: We now demonstrate some properties of the seed sets $\mathcal{S}^g_{ic}$ and $\mathcal{S}^g_{corr}$ for the case of non-identical probabilities. The following three cases of non-identical probabilities were studied (1) Unif$(0, 1)$: $p_{ij}$ drawn i.i.d. from Unif$(0, 1)$; (2) Trivalency: $p_{ij}$ drawn i.i.d. from Unif$\{0.1, 0.01, 0.001\}$; (3) Weighted cascade: $p_{ij} = 1/\deg(i)$, $\deg(i)$ denotes the number of edges incident to $i$. In Table 2 we report the mis-specification ratio under alternate diffusion processes - for the seed set $\mathcal{S}^g_{corr}$ this ratio refers to $f^{ic}(\mathcal{S}^g_{corr})/f^{ic}(\mathcal{S}^g_{ic})$ while for the seed set $\mathcal{S}^g_{ic}$, this ratio refers to $f^{corr}(\mathcal{S}^g_{ic})/f^{corr}(\mathcal{S}^g_{corr})$. We find that both $\mathcal{S}^g_{corr}$ and $\mathcal{S}^g_{ic}$ perform well here under model

| Dataset | Seed Set | p | Mis-spec Ratio | Min Deg($S$) | Average Deg($S$) | Max Deg($S$) | Diam ($S$) |
|---------|----------|---|----------------|--------------|------------------|--------------|------------|
| wikivote | $\mathcal{S}^g_{corr}$ | Unif(0,1) | 0.988 | 41 | 159.475 | 472 | 3 |
| | | Trivalency | 0.928 | 104 | 288.75 | 1167 | 2 |
| | | W.C. | 0.948 | 29 | 217.375 | 537 | 3 |
| | $\mathcal{S}^g_{ic}$ | Unif(0,1) | 0.976 | 12 | 116.85 | 331 | 3 |
| | | Trivalency | 0.908 | 93 | 192.325 | 472 | 2 |
| | | W.C. | 0.949 | 60 | 271.975 | 1167 | 3 |
| polblogs | $\mathcal{S}^g_{corr}$ | Unif(0,1) | 0.981 | 1 | 98.025 | 383 | 6 |
| | | Trivalency | 0.933 | 15 | 159.575 | 467 | 3 |
| | | W.C. | 0.964 | 4 | 140.0 | 467 | 5 |
| | $\mathcal{S}^g_{ic}$ | Unif(0,1) | 0.957 | 1 | 29.825 | 143 | 7 |
| | | Trivalency | 0.928 | 42 | 130.275 | 383 | 3 |
| | | W.C. | 0.96 | 15 | 163.225 | 467 | 4 |

Table 2: Properties of $\mathcal{S}^g_{ic}$ and $\mathcal{S}^g_{corr}$ for non-identical edge probabilities. $k = 40$.

misspecification. However $\mathcal{S}^g_{corr}$ is faster to compute in general as observed earlier. We also report the minimum, maximum and mean degrees of nodes in each of the two seed sets. The diameter of a set refers to the maximum length of the shortest path between all pairs within the set, ignoring the directions of all edges. The sets $\mathcal{S}^g_{corr}$ and $\mathcal{S}^g_{ic}$ are similar in terms of their diameters. The average degree as well as maximum degree of the nodes of $\mathcal{S}^g_{corr}$ is higher than $\mathcal{S}^g_{ic}$ in many cases. For example, for the wikivote dataset, with trivalency, maximum degrees in $\mathcal{S}^g_{ic}$ and $\mathcal{S}^g_{corr}$ are 472 and 1167. In fact the maximum degree of a node in the entire wikivote dataset is 1167. The histograms in Figures 6a and 6b provide more details on polblogs. Figure 5a and Figure 5b illustrate the seed sets $\mathcal{S}^g_{corr}$ and $\mathcal{S}^g_{ic}$, generated on the largest strongly connected component (SCC) of polblogs containing 793 nodes, where an SCC is a subgraph such that every node is connected to every other node. The marginal probabilities **p** are fixed as per the weighted cascade model and $k = 40$.

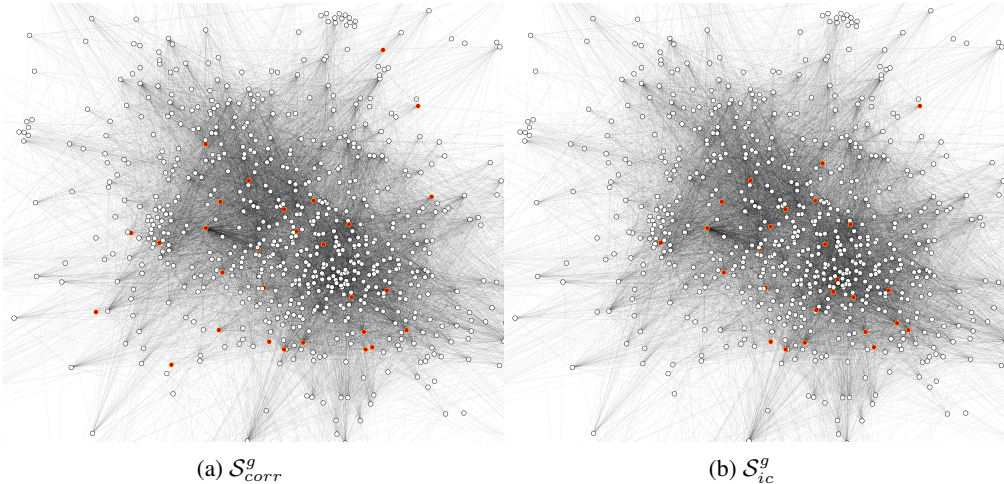

(a) $\mathcal{S}^g_{corr}$           (b) $\mathcal{S}^g_{ic}$

Figure 5: Snapshot of $\mathcal{S}^g_{corr}$ and $\mathcal{S}^g_{ic}$ from the largest strongest connected component of polblogs. The relevant seed nodes are shown in red while all other nodes are shown in white. The marginal probabilities **p** are fixed as per the weighted cascade model and $k = 40$.

**Performance under model misspecification:** In Figure 7 the $f^{corr}(\mathcal{S})$ and the $f^{ic}(\mathcal{S})$ lines represent the expected influence for the set $\mathcal{S}$ under worst case model and IC respectively, for the case of identical probabilities. The range of $f^{ic}(\cdot)$ over 10 runs is at most 0.399% of the mean values and hence these error bars are not visible in the plot. As expected, the curves $f^{ic}(\mathcal{S}^g_{ic})$ and $f^{corr}(\mathcal{S}^g_{corr})$ lie above the curves $f^{ic}(\mathcal{S}^g_{corr})$ and $f^{corr}(\mathcal{S}^g_{ic})$ respectively. The pairs of curves involving $\mathcal{S}^g_{corr}$ are also sandwiched between the pairs involving $\mathcal{S}^g_{ic}$ always. This implies that the loss that arises by using $\mathcal{S}^g_{corr}$ in an IC model is less than that of using $\mathcal{S}^g_{ic}$ under a worst case model.

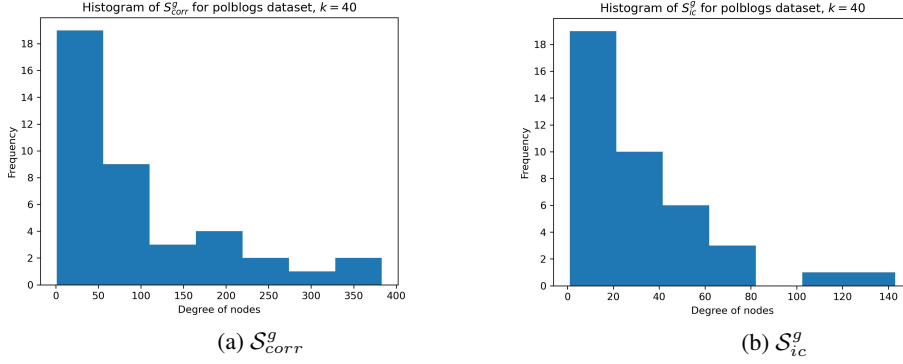

(a) $\mathcal{S}_{corr}^g$
(b) $\mathcal{S}_{ic}^g$

Figure 6: Histogram of degrees of nodes in $\mathcal{S}_{corr}^g$ and $\mathcal{S}_{ic}^g$ on the `polblogs` dataset. Unif$(0, 1)$ for **p**.

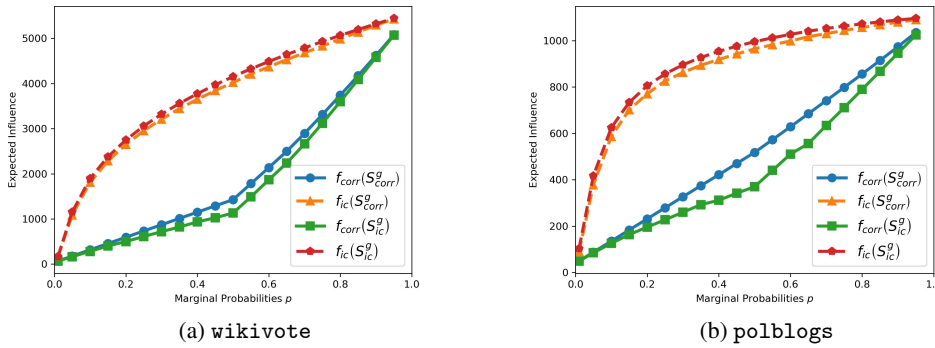

(a) `wikivote`
(b) `polblogs`

Figure 7: Plots of expected influence against identical marginal probabilities, $k = 40$.

## 6 Conclusions and Extensions

We have proposed a model for influence maximization where the activation probabilities of the edges are known, but the joint distribution of these activations is unknown, adversarially chosen upon selection of a seed set. Given a seed set, a polynomial sized linear program can be used to compute this expected influence. The correlation robust function $f^{corr}(\cdot)$ is monotone, submodular and so the greedy algorithm for maximizing $f^{corr}(\cdot)$ holds an approximation guarantee of $1 - 1/e$. For measuring the utility of our model and misspecification under IC, we adapt the price of correlations metric for the influence maximization problem. Using the POC metric, we show instances where using a seed optimal for IC, would hurt the decision maker greatly if an adversarial diffusion process manifests. Finally our experiments provide further insights on real datasets.

Our techniques can be used to deal with the case where a subset $T \subset E$ of the edge activations are known to be mutually independent while dependency information on the rest $(E \setminus T)$ are unavailable. In particular when $|T| \leq \log |E|$, the worst case influence function remains computable in polynomial time. Our techniques can also be used when the probabilities $\mathbb{P}(\tilde{c}_{ij} = 1)$ are only known to lie in an interval $[l_{ij}, r_{ij}]$. Efficient extension to an adaptive model can also follow from our work.

## Broader Impact

The aim of this work is to address the possible pitfalls to the independence assumption in a social network, as used in the study of influence maximization. As discussed previously, how an idea, product, or piece of news makes its way through a network could very well be impacted by natural social biases, thus connecting parts of a social network in ways that could have been unforeseen. The methodology presented thus attempts to make this possibility a consideration during the selection of seed set, and hence find "influential" members to a network regardless of whatever underlying correlations may exist. This potentially can reduce the impact of biases that the independence assumption may cause.

## Acknowledgements

The research of the last three authors was partly supported by the MOE Academic Research Fund Tier 2 grant MOE2019- T2-2-138, "Enhancing Robustness of Networks to Dependence via Optimization".

## Footnotes

[5]Code available at `https://github.com/justanothergithubber/corr-im/`

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
