[Supplementary Material]

# Correlation Robust Influence Maximization: Supplementary Material

**Louis Chen**[*]
Naval Postgraduate School
Monterey, California

**Divya Padmanabhan**[†]
SUTD
Singapore

**Chee Chin Lim**[‡]
SUTD
Singapore

**Karthik Natarajan**[§]
SUTD
Singapore

## Broader Impact

The aim of this work is to address the possible pitfalls to the independence assumption in a social network, as used in the study of influence maximization. As discussed previously, how an idea, product, or piece of news makes its way through a network could very well be impacted by natural social biases, thus connecting parts of a social network in ways that could have been unforeseen. The methodology presented thus attempts to make this possibility a consideration during the selection of seed set, and hence find "influential" members to a network regardless of whatever underlying correlations may exist. This potentially can reduce the impact of biases that the independence assumption may cause.

## The Correlation Robust Influence Function $f^{corr}$

**Theorem 1** *Let $G = (V, E)$ be a directed graph, $\mathcal{S} \subseteq V$ a seed set, and $\mathbf{p} \in [0,1]^E$ a vector of edge likelihoods. Then $\min_{\theta \in \Theta} \mathbf{E}_{\tilde{\mathbf{c}} \sim \theta} [Z(\tilde{\mathbf{c}}, \mathcal{S})]$ is the value to the following polynomial sized linear program.*

$$
\begin{aligned}
\min_{\theta \in \Theta} \mathbf{E}_{\tilde{\mathbf{c}} \sim \theta} [Z(\tilde{\mathbf{c}}, \mathcal{S})] = \min_{\boldsymbol{\pi} \in \mathbb{R}^V} \quad & \sum_{i \in V \setminus \mathcal{S}} \pi_i \\
s.t \quad & \pi_i = 1 \ for \ i \in \mathcal{S}, \\
& \pi_i - \pi_j \leq 1 - p_{ij} \ for \ (i,j) \in E, \\
& 0 \leq \pi_i \leq 1 \ \ for \ i \in V
\end{aligned}
\tag{3}
$$

**Proof:** According to [1], if we let $M$ assume a large value (anything at least $|V \setminus \mathcal{S}|$), then $\min_{\theta \in \Theta(p)} \mathbf{E}_{\tilde{\mathbf{c}} \sim \theta} [Z(\tilde{\mathbf{c}}, \mathcal{S})]$ can be formulated as the following linear program:

---
[*]louis.chen@nps.edu

[†]divya_padmanabhan@sutd.edu.sg

[‡]cheechin_lim@sutd.edu.sg

[§]karthik_natarajan@sutd.edu.sg

$$\min_{\pi \in \mathbb{R}^{V \cup \{s,t\}}, \lambda} \quad \sum_{i,j \in E} 0 \cdot \lambda_{ij}^0 + M \cdot \lambda_{ij}^1 + \sum_{i \in \mathcal{S}} 0 \cdot \lambda_{si}^0 + M \cdot \lambda_{si}^1 + \sum_{i \in N \setminus \mathcal{S}} 0 \cdot \lambda_{it}^0 + \lambda_{it}^1$$

subject to
$$\pi_i - \pi_j \leq \lambda_{ij}^0 \quad \forall (i,j) \in E$$
$$\pi_i - \pi_j - (1 - p_{ij}) \leq \lambda_{ij}^1 \quad \forall (i,j) \in E$$
$$\pi_s - \pi_i \leq \lambda_{si}^0 \quad \forall i \in \mathcal{S}$$
$$\pi_s - \pi_i \leq \lambda_{si}^1 \quad \forall i \in \mathcal{S}$$
$$\pi_i - \pi_t \leq \lambda_{it}^0 \quad \forall i \in V \setminus \mathcal{S}$$
$$\pi_i - \pi_t \leq \lambda_{it}^1 \quad \forall i \in V \setminus \mathcal{S}$$
$$0 \leq \pi \leq 1; \quad \forall i \in V$$
$$\lambda \geq 0$$
$$\pi_s = 1, \pi_t = 0$$

Upon inspection, the program reduces to the desired program. $\qquad \square$

**Corollary 1 (Correlation Robust Influence Likelihood)** *For an arbitrary seed set $\mathcal{S}$ and vector of edge likelihoods $\mathbf{p} \in [0,1]^E$, let $\pi^*$ solve (3). Then for each $i \in V \setminus \mathcal{S}$,*

$$\pi_i^* = \left[ \max_{\gamma \in \Gamma(S,i)} L(\gamma) \right]^+,$$

$$\mathbb{P}_{\theta^*}(\textit{Node } i \textit{ is reachable from } \mathcal{S} \textit{ in } G(\tilde{\mathbf{c}})) = \pi_i^* \qquad \forall \theta^* \in arg \min_{\theta \in \Theta} \mathbf{E}_{\tilde{\mathbf{c}} \sim \theta} \left[ Z(\tilde{\mathbf{c}}, \mathcal{S}) \right].$$

*In light of this, we define the **correlation robust influence likelihood** of $i$ as $\pi_i^*$. In particular, $\pi_i^*$ is no greater than the IC model's likelihood that $i$ is influenced, that is, $\pi_i^* \leq \mathbb{P}_{\theta_{ic}}(\textit{Node } i \textit{ is reachable from } \mathcal{S})$.*

**Proof:** We begin by establishing the equality

$$\pi_i^* = \left[ \max_{\gamma \in \Gamma(S,i)} L(\gamma) \right]^+.$$

Let $i$ be such that $\Gamma(S,i) \neq \emptyset$. Consider any path $\gamma \in \Gamma(S,i)$ and let $\gamma = (i_0 \to i_1 \to i_2 \to \ldots \to i_l = i)$, where $i_0 \in S$. Since $\pi^*$ is feasible to (3), we must have,

$$\pi_{i_0}^* - \pi_{i_1}^* \leq 1 - p_{i_0, i_1}$$
$$\pi_{i_1}^* - \pi_{i_2}^* \leq 1 - p_{i_1, i_2}$$
$$\vdots$$
$$\pi_{i_{l-1}}^* - \pi_i^* \leq 1 - p_{i_{l-1}, i_l}$$

Summation of these inequalities gives $\pi_{i_0}^* - \pi_i^* \leq \sum_{l=1}^{l}(1 - p_{i_{l-1}, i_l})$. Since $i_0 \in S$, it follows that $\pi_{i_0}^* = 1$, so that $\pi_i^* \geq L(\gamma)$. Hence, $\pi_i^* = \left[ \max_{\gamma \in \Gamma(S,i)} L(\gamma) \right]^+$.

On the other hand, observe that if $\Gamma(S,i) \neq \emptyset$, then the decision variable $\pi_i$ has no lower bound other than 0. Further, $\left[ \max_{\gamma \in \Gamma(S,i)} L(\gamma) \right]^+ = 0$, in such a case, as desired.

We next establish the remaining equality

$$\mathbb{P}_{\theta^*}(\text{Node } i \text{ is reachable from } \mathcal{S}) = \pi_i^* \qquad \forall \theta^* \in arg \min_{\theta \in \Theta} \mathbb{E}_{\tilde{\mathbf{c}} \sim \theta} \left[ Z(\tilde{\mathbf{c}}, \mathcal{S}) \right].$$

Taking note of

$$\min_{\theta \in \Theta} \mathbb{E}_\theta[Z(\tilde{\mathbf{c}}, \mathcal{S})] = \min_{\theta \in \Theta} \sum_{i \in V \setminus \mathcal{S}} \mathbb{P}_\theta(\text{Node } i \text{ is reachable from } \mathcal{S} \text{ in } G(\tilde{\mathbf{c}}))$$

$$\geq \sum_{i \in V \setminus \mathcal{S}} \min_{\theta \in \Theta} \mathbb{P}_\theta(\text{Node } i \text{ is reachable from } \mathcal{S} \text{ in } G(\tilde{\mathbf{c}})) \geq \sum_{i \in V \setminus \mathcal{S}} \left[ \max_{\gamma \in \Gamma(\mathcal{S},i)} L(\gamma) \right]^+$$

$$= \sum_{i \in V \setminus \mathcal{S}} \pi_i^* = \min_{\theta \in \Theta} \mathbb{E}_\theta[Z(\tilde{\mathbf{c}}, \mathcal{S})],$$

and that for any $i \in V \setminus \mathcal{S}$, it holds that

$$\min_{\theta \in \Theta} \mathbb{P}_\theta(\text{Node } i \text{ is reachable from } \mathcal{S} \text{ in } G(\tilde{\mathbf{c}})) \geq \left[ \max_{\gamma \in \Gamma(\mathcal{S},i)} L(\gamma) \right]^+,$$

so we arrive at the desired conclusion. $\qquad\square$

**Corollary 2 (Path existence under Correlation Robustness)** *Let $\mathcal{S}$ be an arbitrary seed set, and let $\theta^* \in \Theta$ be any solution to $\min_{\theta \in \Theta} \mathbb{E}_\theta[R(\tilde{\mathbf{c}}, \mathcal{S})]$. Let $\bar{\Gamma}(\mathcal{S}, i) := \arg \max_{\gamma \in \Gamma(\mathcal{S},i)} L(\gamma)$, and $\pi^*$ is any optimal solution to (3). If $i \notin \mathcal{S}$ and $\max_{\gamma \in \Gamma(\mathcal{S},i)} L(\gamma) > 0$, then*

$$\mathbb{P}_{\theta^*}(\cup_{\gamma \in \bar{\Gamma}(\mathcal{S},i)} [G(\tilde{\mathbf{c}}) \text{ contains path } \gamma]) = \pi_i^* = \mathbb{P}_{\theta^*}(\cap_{\gamma \in \bar{\Gamma}(\mathcal{S},i)} [G(\tilde{\mathbf{c}}) \text{ contains path } \gamma]),$$

*In addition, if $\max_{\gamma \in \Gamma(\mathcal{S},i)} L(\gamma) > 0$, then for any path $\gamma \in \bar{\Gamma}(\mathcal{S}, i)$, at most one of the arcs in $\gamma$ is ever missing in the random graph $G(\tilde{\mathbf{c}}) \sim \theta^*$, almost surely.*

**Proof:** If $\theta^*$ solves $\min_{\theta \in \Theta} \mathbb{E}_\theta[R(\tilde{\mathbf{c}}, \mathcal{S})]$, $i \notin \mathcal{S}$, and $\max_{\gamma \in \Gamma(\mathcal{S},i)} L(\gamma) > 0$, then for any $\gamma^* \in \bar{\Gamma}(\mathcal{S}, i)$,

$$\max_{\gamma \in \Gamma(\mathcal{S},i)} L(\gamma) = \mathbb{P}_{\theta^*}(\text{Node } i \text{ is reachable from } \mathcal{S} \text{ in } G(\tilde{\mathbf{c}})) = \mathbb{P}_{\theta^*}(\cup_{\gamma \in \Gamma(\mathcal{S},i)} [G(\tilde{\mathbf{c}}) \text{ contains path } \gamma])$$

$$\geq \mathbb{P}_{\theta^*}([G(\tilde{\mathbf{c}}) \text{ contains path } \gamma^*]) \overset{(5)}{\geq} L(\gamma^*) = \max_{\gamma \in \Gamma(\mathcal{S},i)} L(\gamma).$$

So we conclude that

$$\mathbb{P}_{\theta^*}(\cup_{\gamma \in \Gamma(\mathcal{S},i)} [G(\tilde{\mathbf{c}}) \text{ contains path } \gamma]) = \mathbb{P}_{\theta^*}([G(\tilde{\mathbf{c}}) \text{ contains path } \gamma^*]),$$

which implies

$$\mathbb{P}_{\theta^*}(\cup_{\gamma \in \bar{\Gamma}(\mathcal{S},i)} [G(\tilde{\mathbf{c}}) \text{ contains path } \gamma]) \geq \mathbb{P}_{\theta^*}([G(\tilde{\mathbf{c}}) \text{ contains path } \gamma^*])$$
$$= \mathbb{P}_{\theta^*}(\cup_{\gamma \in \Gamma(\mathcal{S},i)} [G(\tilde{\mathbf{c}}) \text{ contains path } \gamma]),$$

as desired.

For the remaining equality in the statement, we note that if

$$\mathbb{P}_{\theta^*}(\cap_{\gamma \in \bar{\Gamma}(\mathcal{S},i)} [G(\tilde{\mathbf{c}}) \text{ contains path } \gamma]) < \mathbb{P}_{\theta^*}([G(\tilde{\mathbf{c}}) \text{ contains path } \gamma^*]),$$

then $\mathbb{P}_{\theta^*}([G(\tilde{\mathbf{c}}) \text{ contains path } \gamma^*] \setminus [G(\tilde{\mathbf{c}}) \text{ contains path } \gamma']) > 0$ for some $\gamma' \in \bar{\Gamma}(\mathcal{S}, i)$, which means

$$\mathbb{P}_{\theta^*}(\cup_{\gamma \in \Gamma(\mathcal{S},i)} [G(\tilde{\mathbf{c}}) \text{ contains path } \gamma]) \geq \mathbb{P}_{\theta^*}([G(\tilde{\mathbf{c}}) \text{ contains path } \gamma'])$$
$$+ \mathbb{P}_{\theta^*}([G(\tilde{\mathbf{c}}) \text{ contains path } \gamma^*] \setminus [G(\tilde{\mathbf{c}}) \text{ contains path } \gamma'])$$
$$> \mathbb{P}_{\theta^*}([G(\tilde{\mathbf{c}}) \text{ contains path } \gamma^*] \setminus [G(\tilde{\mathbf{c}}) \text{ contains path } \gamma']),$$

a contradiction.

As for the last statement, if $\gamma \in \bar{\Gamma}(\mathcal{S}, i)$, we observe that under the joint distribution $\theta^*$ it cannot be the case that - with positive probability - more than one arc is missing from $G(\tilde{\mathbf{c}})$, else (5) would be a strict inequality, contradicting the fact that Corollary 1 implies that it should be an equality. $\qquad\square$

**Corollary 3** *Given an arbitrary seed set $\mathcal{S}$ and vector of edge likelihoods $\mathbf{p} \in [0,1]^E$, let $\pi^*$ denote the optimal solution to (3). Let $\tilde{q} \sim Unif[0,1]$, $V(\tilde{q}) := \{i : \tilde{q} < \pi_i^*\}$,*

$$E(\tilde{q}) := \{(k,j) : \pi_k^* > \pi_j^*, \tilde{q} \notin [\pi_k^* - 1 + p_{kj}, \pi_k^*]\} \cup \{(k,j) : \pi_k^* \leq \pi_j^*, \tilde{q} \in (0, p_{kj}]\},$$

*and $\mathbf{c}(\tilde{q}) \in \{0,1\}^E$ be such that $c(\tilde{q})_{ij} = 1$ iff $(i,j) \in E(\tilde{q})$. Then $\mathbf{c}(\tilde{q}) \sim \theta^*$ for some $\theta^*$ solving Equation (1). In particular, $V(\tilde{q})$ is the set of all nodes reachable from $\mathcal{S}$ in the graph $G(\tilde{q}) = (V, E(\tilde{q}))$, so that $\mathbb{E}_{\tilde{q}}[|V(\tilde{q})|] = \min_{\theta \in \Theta} \mathbb{E}_{\tilde{\mathbf{c}} \sim \theta}[R(\tilde{\mathbf{c}}, \mathcal{S})] = |\mathcal{S}| + \mathbb{E}_{\tilde{q}}[Z(\mathbf{c}(\tilde{q}), \mathcal{S})].$*

**Proof:** Consider the max-flow problem of $Z(c, \mathcal{S})$ for arbitrary $c \in \{0,1\}^E$. Then the two collections $\{s\} \cup \mathcal{S} \cup \{j : x_{jt}^* = 1, j \in V \setminus \mathcal{S}\}$ and $\{t\} \cup \{j : x_{jt}^* = 0, j \in V \setminus \mathcal{S}\}$ form a minimum $s$-$t$ cut. In particular, $\{j : x_{jt}^* = 1, j \in V \setminus \mathcal{S}\}$ is precisely the set of nodes outside of $\mathcal{S}$ that are reached, and $j$ is reached if and only if the edge $(j, t)$ runs across this minimum cut.

With $\pi^*$ an optimal solution to (3), we may characterize a $\theta^* \in \Theta$ consistent with $\boldsymbol{p}$ that solves $\min_{\theta \in \Theta} \mathbf{E}\left[Z(\tilde{c}, \mathcal{S})\right] = \min_{\theta \in \Theta} \mathbf{E}\left[R(\tilde{c}, \mathcal{S})\right] - |S|$. This characterization will be defined on the probability space $\left((0,1], \mathcal{B}, \lambda\right)$, and for the sake of notation, in the following we'll let $F_{ij}$ denote the cdf for edge $(i, j)$ that is live with probability $\boldsymbol{p}_{ij}$. For all $(i, j) \in E$, if $\pi_i^* > \pi_j^*$, define for all $q \in (0, 1]$,

$$
\tilde{c}_{ij}(q) := \begin{cases} F_{ij}^{-1}(q - \pi_j^*); & \pi_j^* < q \le \pi_i^* \\ F_{ij}^{-1}(1 - p_{ij} + q); & 0 < q \le \pi_i^* - (1 - p_{ij}) \\ F_{ij}^{-1}(1 - p_{ij} - \pi_j^* + q); & \pi_i^* - (1 - p_{ij}) < q \le \pi_j^* \\ F_{ij}^{-1}(q); & \pi_i^* < q \le 1, \end{cases}
$$

otherwise if $\pi_i^* \le \pi_j^*$ define $\tilde{c}_{ij}(q) := F_{ij}^{-1}(1 - q)$. Finally, for all $(i, j) \notin E$ but are auxillary arcs with $s$ or $t$ as an endpoint, we can let $\tilde{c}_{ij}(q) := +\infty$ if $i = s$, else $\tilde{c}_{ij}(q) := 1$ for the case that $j = t$. As well, we define

$$
\tilde{\chi}_{ij}(q) := \begin{cases} 1; & \pi_i^* > \pi_j^*, q \in [\pi_j^*, \pi_i^*] \\ 0; & \text{otherwise.} \end{cases}
$$

The resulting random vector $\tilde{c}$ has as its distribution a solution to $\min_{\theta \in \Theta} \mathbf{E}\left[Z(\tilde{c}, \mathcal{S})\right]$. This follows after adopting the arguments in Theorem 3.1 of [1]. It is not hard to see that with $\tilde{q} \sim Unif(0, 1]$, $E(\tilde{q})$ as defined in the statement is precisely $\{(k, j) : \tilde{c}_{kj}(\tilde{q}) = 1\}$. Furthermore, according to Theorem 3.1 of [1], $\tilde{\chi}_{jt}(\tilde{q})$ is 1 if and only if $(j, t)$ runs across the minimum cut - equivalently, when $j$ is reached. And since $\pi_t^* = 0$ always, we arrive at the characterization of $V(\tilde{q})$. $\qquad\square$

## Correlation Robustness: Maximization and Robust Ratios

**Theorem 2** *The problem of computing $\max_{\mathcal{S}:|\mathcal{S}|\le k} f^{corr}(\mathcal{S})$, given a graph $G = (V, E)$, a vector of edge likelihoods $\mathbf{p} \in [0, 1]^E$, and an integer number $k$, is NP-Hard. In particular, we have the following exact formulation as a mixed-integer program.*

$$
\max_{\mathcal{S}:|\mathcal{S}|\le k} f_p^{corr}(\mathcal{S}) = max \sum_{(i,j)\in E} z_{ij}(p_{ij} - 1) + \sum_{i \in V} w_i
$$

$$
1 - y_i - \sum_{j:(j,i)\in E} z_{ji} + \sum_{j:(i,j)\in E} z_{ij} \ge 0 \; \forall i \in V
$$

$$
w_i \ge |V| x_i + y_i - |V| \; \forall i \in V
$$

$$
w_i \le \min(|V| x_i, y_i) \; \forall i \in V
$$

$$
\sum_{i \in V} x_i = k
$$

$$
y_i \ge 0 \; w_i \ge 0 \; \forall i \in V
$$

$$
z_{ij} \ge 0 \; \forall (i, j) \in E
$$

$$
x_i \in \{0, 1\} \; \forall i \in V
$$

**Proof:** We prove the hardness of computing $\max_{\mathcal{S}:|\mathcal{S}|\le k} f^{corr}(\mathcal{S})$ through a reduction from the set cover problem. The proof is along the lines of the proof of hardness of the independent cascade model in [2]. In the set cover problem, there is a universe of elements $\Omega = \{1, \ldots, n\}$, a collection of subsets $J_1, \ldots, J_m \subseteq \Omega$ (whose union gives $\Omega$), and an integer $k$. The decision version of the set cover problem is to check if there exists a collection of $k$ subsets, whose union gives $\Omega$. We will now reduce an instance of set cover problem to (2). For this, consider a bipartite graph with a total of $m + n$ vertices corresponding to the $m$ subsets and the $n$ elements of $\Omega$. This bipartite graph contains an edge between a subset node $i$ and an element node $j$ if $j \in J_i$. Fix $p_{ij} = 1$ for all edges $(i, j)$ in this graph. Then there exist $k$ subsets whose union is $\Omega$ is and only if the optimal value to (2) is $k + n$.

Next we will derive the MILP formulation. Using Theorem 1, we have,

$$\max_{\mathcal{S}:|\mathcal{S}|\leq k} f_p^{corr}(\mathcal{S}) = \max_{x\in\mathbb{R}^V}\min_{\pi\in\mathbb{R}^V} \quad \sum_i \pi_i$$

$$\text{subject to} \quad x_i \leq \pi_i \ \ \forall i \in V$$
$$\pi_i - \pi_j \leq 1 - p_{ij} \ \ \forall (i,j) \in E$$
$$0 \leq \pi_i \leq 1 \quad \forall i \in V$$
$$\sum_{i\in V} x_i = k$$
$$x_i \in \{0,1\} \quad \forall i \in V$$

The dual of the inner minimization problem is,

$$\max_{\mathbf{z}\geq 0,\mathbf{y}\geq 0,\mathbf{w}\geq 0} \sum_{(i,j)\in E} z_{ij}(p_{ij}-1) + \sum_{i\in V} x_i y_i : 1 - y_i - \sum_{j:(j,i)\in E} z_{ji} + \sum_{j:(i,j)\in E} z_{ij} \geq 0 \ \forall i \in V$$

Further we linearize the product terms $w_i = x_i y_i$. Summing up the inequality over all $i$ gives us, $\sum_{i\in V}(1 - y_i - \sum_{j:(j,i)\in E} z_{ji} + \sum_{j:(i,j)\in E} z_{ij}) \geq 0$. The terms involving $z$ cancel out and we are left with $\sum_{i\in V} y_i \leq V$ and since $y_i \geq 0$ for all $i$, we get an upper bound $y_i \leq V$.

Using the bounds $0 \leq x_i \leq 1$ and $0 \leq y_i \leq |V|$, the McCormick inequalities introduced in [3] for $w_i$ give us,

$$|V|x_i + y_i - |V| \leq w_i \leq \min(|V|x_i, y_i) \ \forall i \in V$$

To see that these inequalities are sufficient to capture $w_i = x_i y_i$, when $x_i \in \{0,1\}$, first let $x_i = 0$. Then the inequalities give us $y_i - |V| \leq w_i \leq \min(0, y_i)$ and along with the fact that $w_i \geq 0$, we get $w_i = 0$. Now Let $x_i = 1$. Then the inequalities give us $y_i \leq w_i \leq \min(|V|, y_i) = y_i$. Therefore we get $w_i = y_i$ and hence these inequalities are tight. $\qquad \square$

**Theorem 3** *The correlation robust influence function $f^{corr} : 2^V \to \mathbb{R}_+$ is a monotone, submodular function.*

**Proof:** Since $f^{corr}(\mathcal{S}) = |\mathcal{S}| + \min_{\theta\in\Theta} \mathbb{E}_\theta[Z(\tilde{\mathbf{c}}, \mathcal{S})]$, submodularity of $g(\mathcal{S}) := \min_{\theta\in\Theta} \mathbf{E}_{\tilde{\mathbf{c}}\sim\theta} Z[\tilde{\mathbf{c}}, \mathcal{S}]$ implies submodularity of $f^{corr}$. If two seed sets $S$ and $T$ with $S \subset T$ and vertex $v \notin T$ are given, then by (7),

$$g(S+v) - g(S) = \sum_{i\notin(S\cup v)} \max\left( [\max_{\gamma\in\Gamma(S,i)} L(\gamma)]^+, [\max_{\gamma\in\Gamma(\{v\},i)} L(\gamma)]^+ \right)$$

$$- \left[ \sum_{i\notin(S+v)} [\max_{\gamma\in\Gamma(S,i)} L(\gamma)]^+ + [\max_{\gamma\in\Gamma(S,v)} L(\gamma)]^+ \right]$$

$$= \sum_{i\notin(S+v)} \left[ [\max_{\gamma\in\Gamma(\{v\},i)} L(\gamma)]^+ - [\max_{\gamma\in\Gamma(S,i)} L(\gamma)]^+ \right]^+ - [\max_{\gamma\in\Gamma(S,v)} L(\gamma)]^+ \quad (1)$$

$$\geq \sum_{i\notin(T+v)} \left[ [\max_{\gamma\in\Gamma(\{v\},i)} L(\gamma)]^+ - [\max_{\gamma\in\Gamma(T,i)} L(\gamma)]^+ \right]^+ - [\max_{\gamma\in\Gamma(T,v)} L(\gamma)]^+$$

$$= g(T+v) - g(T),$$

as desired. As for monotonicity, simply observe that by (1),

$$f^{corr}(S+v) - f^{corr}(S) = g(S+v) - g(S) + 1 \geq 1 - [\max_{\gamma\in\Gamma(S,v)} L(\gamma)]^+ \geq 0.$$

$\qquad \square$

**Corollary 4** *Let $\mathcal{S}_{corr}^g$ denote the seed set generated upon termination of the greedy algorithm for maximization of $f^{corr}$. Then*

$$f^{corr}(\mathcal{S}_{corr}^g) \geq (1 - 1/e) \max_{|\mathcal{S}|\leq k} f^{corr}(\mathcal{S})$$

**Proof:** By Theorem 3 and known approximation guarantees for submodular optimization [4] we get the result. □

**Computations for Example 2, POC study**

Figure 1: Example 2 for POC study

We consider the tree in Figure 1 with a root node, containing $l$ children. There are a total of $l$ paths from the root to all the leaf nodes, starting from the root node. Each path contains $m + 2$ nodes (apart from the root). The labels on the nodes indicate the "type" of each node. Between nodes of type $0$ and $1$ as well as between type $1$ and type $2$ nodes, the activation probability $= 0.5$. For all other edges, activation probability is $1$. The total number of nodes in the graph is $n = l(m + 2) + 1$. Suppose we are interested in choosing a single seed node, so $k = 1$.

*Independent cascade model*: We first compute the values of $f^{ic}(.)$ for each type of node.

Type 2: For such nodes, $f^{ic}(\{2\}) = m + 1$. Also it can be verified that nodes of type 2 reach more than nodes of type 3, 4, ... $m + 2$.

Type 1: There is one random edge which, if active, will enable $m + 1$ nodes to be reached. However if this edge is inactive, none of the nodes are reached. Therefore, $f^{ic}(\{1\}) = \frac{m+1}{2} + 1$.

Type 0 (root): Here we are $l$ sub-trees (each corresponding to a path graph) in which the nodes could be potentially reached. Let the number of nodes reached in each of the sub-trees be denoted by the random variables $\tilde{X}_1, \ldots, \tilde{X}_l$. The object of our interest is $\mathbb{E}_{\theta_{ic}} \left[ \sum_{i=1}^{l} \tilde{X}_i \right] + 1$. $\tilde{X}_i$ takes values $m + 2, 1$ and $0$ with probabilities $0.25, 0.25$ and $0.5$ respectively. and therefore $\mathbb{E}[\tilde{X}_i] = (m + 3)/4$. Therefore the overall reachability $f^{ic}(\{0\}) = 1 + l(m + 3)/4$.

Clearly the choice to be made is between the root node and any node of type 2 (as node 2 is always better than node 1 (assuming $m \geq 1$). The root node is preferred when $l(m + 3)/4 \geq m$ which occurs when $l \geq \frac{4m}{m+3}$.

*Worst case analysis*: We perform a similar analysis on the values of $f^{corr}(\cdot)$ too. For any type 2 node, we have $f^{corr}(\{2\}) = m + 1$. When $\mathcal{S} = \{1\}$, $f^{corr}(\{1\}) = 1 + \frac{m+1}{2}$ as an optimal solution to the LP that computes $f^{corr}(\{1\})$ is $\pi_2^* = \pi_3^* = \ldots = \pi_{m+2}^* = 0.5$ from Corollary 1.

Type 0 (root): In each sub-tree of the root node, our LP solution gives $\pi_1^* = 0.5, \pi_2^* = \pi_3^* = \ldots = \pi_{m+2}^* = 0$. Therefore $f^{corr}(\{0\}) = 1 + l/2$.

Between type 0 and type 2 nodes, type 0 is selected whenever $l > 2m$ and a type 2 node can be selected otherwise.

Suppose $\frac{4m}{m+3} \leq l \leq 2m$. Then if $k = 1$, $\mathcal{S}_{corr}$ is any one of the type 2 nodes while $\mathcal{S}_{ic} = \{0\}$. Then the price of correlations is $\frac{(l/2)+1}{m+1}$. If $l = \frac{4m}{m+3}$, then POC $= \frac{2m+3}{(m+1)(m+3)}$ which tends to zero as $m \to \infty$.