[Reviews · NeurIPS 2020]

Review 1

Summary and Contributions: ### Update ### I thank the authors very much for kindly answering my questions. As regards the technical novelty and corollaries, my concerns have been mostly resolved. I would appreciate it if the authors could improve the manuscript so that those parts become clearer. Concerning the practical impact, what the authors argue would be probably true. However, the presented experimental studies do not seem to be convincing enough. Considering the above things, I slightly raise my score. ############# The paper proposes a correlation-robust method for influence maximization, where live-edge probabilities can be correlated as long as they are consistent with certain marginals. The objective function value can be computed via polynomial-size LP, and it is also monotone and submodular. Consequently, the greedy algorithm yields 1-1/e approximation guarantee. A motivative example with a small "price of correlation" value is presented. Experiments compare the proposed method with the one that ignores the correlation (IC model).

Strengths: - The paper considers a novel problem setting and obtains good theoretical results. - Unlike the IC model case, the proposed method can exactly compute the objective value, which yields a computational advantage of the proposed method.

Weaknesses: - In terms of technical contributions, it seems that the most important part, Theorem 1, relies heavily on [Chen et al. 2020]. Considering this, the overall technical novelty appears to be weak. - In the experiments, the advantages of the proposed method do not seem to be significant, which raises a question: Are there practical situations where ignorance of correlation results in a serious failure? While an artificial example is shown in Example 2, the proposed method should be motivated more with practical scenarios.

Correctness: The claims and methods seem to be correct, although I did not check the proofs.

Clarity: The structure of the paper seems to have room for improvement. For example, it was hard to understand the purpose of the discussion related to Corollaries 1--3. In my understanding, once Theorem 1 is obtained, we can perform the greedy algorithm; thus the goal of the paper is achieved (by showing the monotonicity and submodularity as in Theorem 3). These can be explained more concisely without Corollaries 1--3. The remaining part of Section 3 is interesting but appears to digress from the main topic.

Relation to Prior Work: It seems to be enough, although I'm not familiar with this research area.

Reproducibility: No

Additional Feedback:


Review 2

Summary and Contributions: The paper studies a problem of influence maximization in which the activations are not chosen probabilistically, but in an adversarial manner, which can lead to correlations between edges. The authors show that the influence for a given seed set can be computed by a linear program, and this function is submodular. The formulation is evaluated on two networks, and shows differences between the correlated and independent cascades

Strengths: The setting of robust optimization under an adversarial setting is interesting, and well motivated. The paper is technically interesting

Weaknesses: The motivation for this specific model is not very clear from the perspective of network science and diffusion processes, as discussed below. The model is motivated from an adversarial perspective, but there are other models, which seem more natural. The model seems to be a force fit to a paper on distributionally robust max flow (ref [1]), where such a model is more natural. The presentation is very poor, and lots of notation is used inconsistently, or not defined properly

Correctness: This is hard to completely verify do due to the notation and presentation

Clarity: Poor

Relation to Prior Work: There are important papers on variations of influence maximization and the objectives, which are missing

Reproducibility: Yes

Additional Feedback: There is a lot of work on diffusion models with correlations. An example is the complex contagion models, e.g., (Duncan J Watts. A simple model of global cascades on random networks. Proceedings of the National Academy of Sciences USA, 99:5766–5771, 2002). Such models do not assume the live edges are independent. None of this line of work is discussed in the related work section. In contrast, some of the papers mentioned in Section 2, e.g., ref [8] is on a different objective for influence, namely, the probability of a large influence, instead of the expectation. So its not clear that is relevant here. Section 3: the definition of the set \Theta is confusing. What is \theta(c)? If it is also a vector, what is \mathbf{p_{ij}}? It looks like \mathbf{p} is a vector and p_{ij} is the component corresponding to edge (i, j). In in the definition of f^{corr}(S) in equation (1), the expectation seems to be over \theta? But on page 1, the expectation is over c. The notation is slightly different in the supplementary file Page 7: the heuristic mentioned, using accelerated greedy for computing S_{corr} is not clear here. For instance, ref [19] mentioned here is defined for the IC model. More details are needed (this is not described in the Supplementary info either) While the results show that using S_{corr} in an IC model is much better than using S_{ic} under correlations, it would be good to make a more substantial statement than just a qualitative statement through two plots --------------------------- I have read the author feedback and feel they have addressed many of the concerns. I still feel the presentation needs to be improved


Review 3

Summary and Contributions: In this paper, the authors study the problem robust influence maximization under the adversarial setting of worst correlation between edge activation given fixed marginal edge activation probabilities. The authors show that first the worse correlation can be solved via a polynomial size Linear programming. The proof is via reduction to randomized max flow problem. Moreover, the authors provide exact characterization of the worst correlation via activation path and node activation probability. Moreover, the authors prove that the objective function of the robust influence maximization problem is submodular and be solve via greedy algorithm. The authors also provide example graph for the Price of Correlation. Finally, empirical experiment is carried out on several real-world social networks. I have read the authors's feedback and will remain my original evaluation.

Strengths: 1. The authors study an interesting problem on robust influence maximization. The violation of independent assumption is another interesting problem to study besides the perturbation of edge probabilities. 2. The paper provides strong and complete theoretical results. The authors first prove the polynomial solvability for the worst correlation and also the submodularity of the robust influence maximization problem. 3. The authors provide insightful characterization of the worst correlation solution via live-path and node activation probability. It provides interesting distinction between IC model and IC model under worst correlation perturbation.

Weaknesses: 1. For practical problem, the power of the adversarial is limited. It would be interesting to study the problem where the adversarial can only perturb the correlation but only up to an extent. It would provide more practical application of the method. 2. It would be interesting for the authors to provide more empirical characterization of the seed sets selected by IC and the worst correlation. Some visualization of the seed under small networks would be very interesting.

Correctness: The paper is technically sound with detailed proofs and valid empirical experiment.

Clarity: The paper is well written and easy to follow.

Relation to Prior Work: Yes, the paper provides thorough review of related work.

Reproducibility: Yes

Additional Feedback:


Review 4

Summary and Contributions: The authors address the influence maximization problem from the distributionally robust point of view which is a major analysis aspect of the recent eight years. They formulate a new model in which they allow correlations. In their setting the computations are still efficient therefore the optimization problem has an approximation guarantee. This allows for efficient evaluation of the influence in contrast to the ground model ([16]). They also connect the correlation-robust model to the linear threshold model.

Strengths: - Clarity of ideas, both a theoretical founding and an applied evaluation of the model. - The preliminary assumptions required for the distributions are the marginal probabilities (this is more plausible for a learning setting). - The experiments show that the seed sets obtained the authors' model has a more expanded 'configuration' in the input graph (in comparison to the IC model). - The authors employ Price of Correlation (POC) to contrast independence to correlations and this leads to interesting results. - It seems that this model is a model different from that of [14]. Here the authors allow for correlations from adversaries. - Figures 1,2,3 are instructive. - This work is Important to the social networks audience . AFTER FEEDBACK The authors addressed most comments made by the reviewers. The typos are planned to be corrected, so the paper will be better presented. Its value seems high enough to be accepted.

Weaknesses: - Line 54: Is 'interpretable' program relevant to the notion described in the work of 'Doshi-Velez, F., & Kim, B. (2017). Towards a rigorous science of interpretable machine learning. arXiv preprint arXiv:1702.08608'? - Line 19, 37, 39: A reference for the 'Influence maximization' problem may be provided. The distribution may be more formally given (e.g. which p_{ij} sum to 1). To be able to refer to the joint distributions, there should be a more concrete statement of the p_{ij}. Or maybe a preamble of line 103. - Line 52: Some more details about the polynomial time character of the formulation may clarify your statement about the LP. - Line 103: The strategy space of the adversary implied in the equation is strongly pessimistic (why consider all possible correlations?). This can be used in a follow up work. It seems that it does not reduce the value of the current model.

Correctness: The theoretical model seems concise and supported enough by the theorems and lemmas. The experiments verify the model in an applied way and demonstrate its soundness. Specifically, they show the computational efficiency promised by the theoretical model is indeed higher than the one of IC model of literature.

Clarity: The paper has a succinct description of the model. The text is meticulously written. It introduces the reader to the key parts without too many details (e.g. the Related Work part is not overwhelmingly big although the problem of IC is widely studied).

Relation to Prior Work: - This work is different from the IC model described in the seminal work of reference [16]. The presented model drops the independence assumption for the 'liveness' of the edges (as noted in the 3rd contribution) as well as the adapted variant of IC. Thus it creates a less simplistic setting and so seems to be novel. - POC is used to contrast independence to correlations and this leads to interesting results.

Reproducibility: Yes

Additional Feedback: The authors may include one short comment on follow up work emerging directly from the current paper (e.g. adaptive influence optimization version of the problem). The broader impact of this work is addressed in the Introduction (transmission of information in graphs and social networks, diffusion of medical innovations). More information in that is encouraged (e.g. in the Conclusions section), but not absolutely needed as a separate section.

[Author Response · NeurIPS 2020]

We thank all the reviewers for the detailed perusal and valuable suggestions. Please find our responses below.

**Reviewer 1** *On technical novelty:* On observing the role of max flows in modeling diffusion processes, we took
inspiration from Chen et al [2020]'s study and formulated a new robust influence maximization model. Our first step in
establishing the tractability of the model's influence function $f^{corr}$ (Theorem 1) involves adapting a key result in Chen
et al [2020], but our study as a whole concerns much more. Towards optimizing $f^{corr}$, we show it is a submodular
function. This does not follow from the literature; indeed, if $g(\cdot)$ and $h(\cdot)$ are submodular set functions, the pointwise
minimum $\min(g(\cdot), h(\cdot))$ need not be a submodular function in general (see the book on submodular functions by
Bach, 2013). But in our case of $f^{corr}(\mathcal{S}) = \min_{\theta \in \Theta} \mathbb{E}_\theta[R(\tilde{\mathbf{c}}, \mathcal{S})]$ as a pointwise minimum of submodular functions
$\mathbb{E}_\theta[R(\tilde{\mathbf{c}}, \mathcal{S})]$ over $\theta \in \Theta$, it is submodular. Beyond study of the optimization problem, our robust model enables a study
of how costly the independence assumption in IC can be (Sec 4.2). Further advantages follow from Corollaries 1-3.

*On Corollaries 1-3:* Cor 1 reveals several distinguishing features of our model: (a) in contrast to IC, we can efficiently
compute the (worst-case) node activation probability for any node i via $\pi_i^*$; (b) with $f^{corr}(\mathcal{S}) = \sum_{i \in V \setminus \mathcal{S}} \pi_i^*$, we find
that the adversary's problem is additively separable- "the worst-case sum is the sum of the worst-cases" (see Eqn (6));
(c) there is a simpler computation for $f^{corr}$ (using shortest path computations instead of the LP, see line 219 on page 7).
Cor 2 reveals that only a subset of the paths from $\mathcal{S}$ to node $i$ propagate influence to $i$, unlike in IC where all paths
contribute. Cor 3 describes how to simulate $\theta^*$ and makes an interesting comparison with LTM - a surprising byproduct
of our original aim for a robust treatment of IC. Since all these pertain to $f^{corr}$ they were included in Sec 3.

*Scenarios, broader impact:* While a graph captures observable network connections, there may be latent variables
causing segments of a network to exhibit correlation in propagating influence. For example, in spreading news articles
on Twitter, each article resonates with members in complex, correlated ways due to hidden variables (like political party
affiliation, demographics), so our work finds those members whose influence is robust to the choice of article promoted.

**Reviewer 2** *On motivation and other models,references:* There is a vast literature on network models, and IC
is particularly well-studied in influence maximization. Our interest was in examining the classical independence
assumption by formulating a model most antagonistic to IC in that the diffusion process is governed by an adversary
(for motivation and broader impact, please refer to the para above). For references, we chose the ones most relevant to
IC and its robust treatments, due to space. We will be happy to cite more works, including those suggested.

*On accelerated greedy:* The accelerated greedy technique [23] can be used to maximize any monotone submodular
function and only requires an oracle that evaluates the function. So we use it to maximize $f^{corr}$; we will provide details.

*On notations:* In Sec 3, $\theta$ is a distribution over $2^{|E|}$ graph realizations, where any fixed graph realization is a vector
$\mathbf{c} \in \{0,1\}^{|E|}$. $\theta(\mathbf{c})$ denotes the probability mass assigned to $\mathbf{c}$. We denote the underlying random variable as $\tilde{\mathbf{c}}$.
Expectations are always taken over $\tilde{\mathbf{c}}$ with pmf $\theta$ (written $\tilde{\mathbf{c}} \sim \theta$). $\mathbf{p} = (p_e)_{e \in E}$ is a vector with $|E|$ components where
$p_e$, equivalently $p_{ij}$, denotes the activation probability of edge $e = (i,j) \in E$. All of this should clarify the typo on line
103, for which we apologize.

*Beyond qualitative statement through plots:* The POC in Sec 4.2 provides a technical treatment to compare $\mathcal{S}_{ic}$ and
$\mathcal{S}_{corr}$ under correlations. As the comparison is graph-dependent, a completely general conclusion cannot be drawn.

**Reviewer 3** *On extensions:* Our techniques can handle the case where a subset $T \subset E$ of the edge activations are
known to be mutually independent while dependency information on the rest $(E \setminus T)$ are unavailable. In particular
when $|T| \leq \log|E|$, the worst case influence function remains computable in polynomial time. If the adversary is
additionally allowed to choose $p_{ij}$ within $[a_{ij}, b_{ij}]$, the problem remains tractable. In general tractability under other
ambiguity sets $\Theta$ is not guaranteed (e.g., further constraining the correlation structure can make it NP-hard to check
non-emptiness of $\Theta$, see Georgakopoulos et al, Probabilistic Satisfiability, 1988); we will need a separate investigation
towards these aspects. Efficient extension to an adaptive model can also follow from our work. Our paper may be
viewed as a start to this line of work in robust influence maximization. We will discuss the above extensions.

*Visualization:* Since the datasets used are too large to visualize the seed sets, we have reported some properties of these
seed-sets. While small examples have been provided in Section 4.2, we will provide a few snapshots of larger examples.

**Reviewer 4** *Interpretability:* The variables in our LP have meaningful interpretations as $\pi_i^*$ can be viewed as the worst
case probability that node $i$ is reachable from a seed set $\mathcal{S}$ (Corollary 1). The overall optimization problem thus is
meaningful and conforms to requirements in Doshi-Velez et al, 2017.

*Polynomial character of LP:* Our LP in Theorem 1 being of efficient size (with $|V|$ variables and $k + |V| + |E| \leq$
$2|V| + |E|$ linear constraints) can be solved in time polynomial in the inputs $|E|$ and $|V|$.

*On follow up work:* Please refer to our answer for reviewer 3 'On extensions'. We will also add more on broader impact.

*Line 19, 37, 39, 103:* We will add references and preamble to line 103 based on our answer 'On notations' to reviewer 2.

[Meta-Review · NeurIPS 2020]

Reviewers appreciate the authors' rebuttal and have a thorough discussion. All are in favor and believe the paper has solid theoretical results. One reviewer points out the paper could include a more comprehensive literature review, as the model is closely related to some existing ones but this was not clearly explained in the paper.